# The cytosolic N-terminal region of heterologously-expressed transmembrane channel-like protein 1 (TMC1) can be cleaved in HEK293 cells

**Soichiro Yamaguchi**[1]*, **Maho Kamino**[2], **Maho Hamamura**[2], **Ken-ichi Otsuguro**[3]

**1** Laboratory of Physiology, Department of Basic Veterinary Sciences, Faculty of Veterinary Medicine, Hokkaido University, Sapporo, Hokkaido, Japan, **2** Laboratory of Pharmacology, Department of Basic Veterinary Sciences, School of Veterinary Medicine, Hokkaido University, Sapporo, Hokkaido, Japan, **3** Laboratory of Pharmacology, Department of Basic Veterinary Sciences, Faculty of Veterinary Medicine, Hokkaido University, Sapporo, Hokkaido, Japan

* souya@vetmed.hokudai.ac.jp

**Data Availability Statement:** All relevant data are within the paper and its Supporting information files.

## Abstract

Transmembrane channel-like protein 1 (TMC1) is a transmembrane protein forming mechano-electrical transduction (MET) channel, which transduces mechanical stimuli into electrical signals at the top of stereocilia of hair cells in the inner ear. As an unexpected phenomenon, we found that the cytosolic N-terminal (Nt) region of heterologously-expressed mouse TMC1 (mTMC1) was localized in nuclei of a small population of the transfected HEK293 cells. This raised the possibility that the Nt region of heterologously-expressed mTMC1 was cleaved and transported into the nucleus. To confirm the cleavage, we performed western blot analyses. The results revealed that at least a fragment of the Nt region was produced from heterologously-expressed mTMC1. Site-directed mutagenesis experiments identified amino acid residues which were required to produce the fragment. The accumulation of the heterologously-expressed Nt fragment into the nuclei depended on nuclear localization signals within the Nt region. Furthermore, a structural comparison showed a similarity between the Nt region of mTMC1 and basic region leucine zipper (bZIP) transcription factors. However, transcriptome analyses using a next-generation sequencer showed that the heterologously-expression of the Nt fragment of mTMC1 hardly altered expression levels of genes. Although it is still unknown what is the precise mechanism and the physiological significance of this cleavage, these results showed that the cytosolic Nt region of heterologously-expressed mTMC1 could be cleaved in HEK293 cells. Therefore, it should be taken into account that the cleavage of Nt region might influence the functional analysis of TMC1 by the heterologous-expression system using HEK293 cells.

## Introduction

Sound-evoked vibration is detected by hair cells in the cochlea, and head motion and position relative to gravity are detected by hair cells in the vestibule [1]. Mechanoelectrical transduction

**Funding:** This research was funded by JSPS KAKENHI Grant Number 16K08066, THE AKIYAMA LIFE SCIENCE FOUNDATION, Takeda Science Foundation, and The Uehara Memorial Foundation to S.Y. This work was the result of using research equipment shared in MEXT Project for promoting public utilization of advanced research infrastructure (Program for supporting introduction of the new sharing system) Grant Number JPMXS0420100619. The funders had no role in study design, data collection and analysis, decision to publish, or preparation of the manuscript.

**Competing interests:** The authors have declared that no competing interests exist.

(MET) channel converts such mechanical stimuli into electrical signals at the top of the stereocilia of hair cells [1]. When stereocilia are deflected, MET channels open, causing influx of $K^+$ and $Ca^{2+}$, and depolarizes the membrane potential. As the molecular identity of the channel, transmembrane channel-like 1 (TMC1) is becoming widely accepted as a pore-forming subunit of MET channel, along with TMC2 [2, 3]. Not only vertebrates but also metazoa possess TMC-like genes [4].

In order to identify functions of proteins, heterologous expression systems are usually a powerful tool for analyses of ion channel functions. Regarding TMC1, it has been shown that heterologously-expressed TMC1 retains in endoplasmic reticulum (ER) and fails to traffic to the plasma membrane in various heterologous cells [2, 5, 6]. However, it was reported that coexpression of KCNQ1 (Potassium Voltage-Gated Channel Subfamily Q Member 1) rescued expression of TMC1 in the plasma membrane of CHO cells although mechanosensitive currents were still not measured from the cells. Therefore, functional analyses of TMC1 in heterologous expression systems may be possible soon if additional binding partners required for the functional expression of TMC1 are revealed.

Predicted TMC1 model structures showed a dimeric assembly, with each subunit containing cytosolic N-terminus, ten transmembrane regions, and cytosolic C-terminus [2, 7, 8]. Its N-terminal (Nt) region is longer than C-terminal (Ct) region (Nt: ~185 amino acid residues, Ct: ~40 amino acid residues) and contains a characteristic sequence: five alternating clusters of acidic amino acid residues and basic amino acid residues within the first 80 amino acid residues, which are not conserved in other TMCs (TMC2-TMC8, widely expressed in many species [4]) (Fig 1A). The region beneath the first transmembrane region was suggested to be involved in ER retention of TMC1 [9] and the Nt domain (amino acids 81–130) of mouse TMC1 (mTMC1) was reported to be a binding cite of $Ca^{2+}$- and integrin-binding protein 2 (CIB2) [5, 10] although the other binding cite of CIB2 in the first cytosolic loop of mTMC1 was also reported [11]. However, despite its characteristic amino acid sequence, the intrinsic function of the Nt region of TMC1 has not been elucidated.

Although many transmembrane proteins function as receptors, ion channels, transporters, and enzymes, another example of their function is transcriptional regulation. The transcriptional regulations by transmembrane proteins are mediated by the cytosolic Nt and Ct regions, which are cleaved and released from the membrane proteins, such as SREBP (sterol regulatory element-binding protein), ATF-6 (Activating transcription factor 6), and Notch [12]. Such transcriptional regulations by cleaved and released cytosolic fragments have been also reported as a function of some ion channels and ion channel-related membrane proteins. For example, the cytosolic Ct region of polycystin-1, which form a cation channel complex with polycystin-2 [13], is cleaved by proteolysis and transported into the nuclei, and regulates transcription by activating STAT [14–16]. Moreover, the Ct regions of transient receptor potential melastatin 6 (TRPM6) and 7 (TRPM7), which are nonselective cation channels, were also reported to be cleaved and regulate transcription through histone phosphorylation by their kinase domains within the Ct regions [17–20].

The cleaved fragments of SREBP, Notch, and polycystin-1 possess nuclear localization signals (NLSs) [12, 15], which act as a signal fragment that mediates the transport of proteins from the cytoplasm into the nucleus [21]. One type of NLSs is monopartite classical NLS (cNLS, K-K/R-X-K/R [22]). Importin α binds to cNLS and transports the proteins which possess cNLS into the nuclei in conjunction with importin β [22]. The Nt regions of mTMC1 and human TMC1 (hTMC1) possess putative two overlapped monopartite cNLSs (KRKRTR: KRKR and KRTR are cNLSs.) (Fig 1A). However, it has not been evaluated whether the putative cNLSs function as NLS, and physiological meaning of the presence of cNLSs in Nt region of TMC1s are not understood.

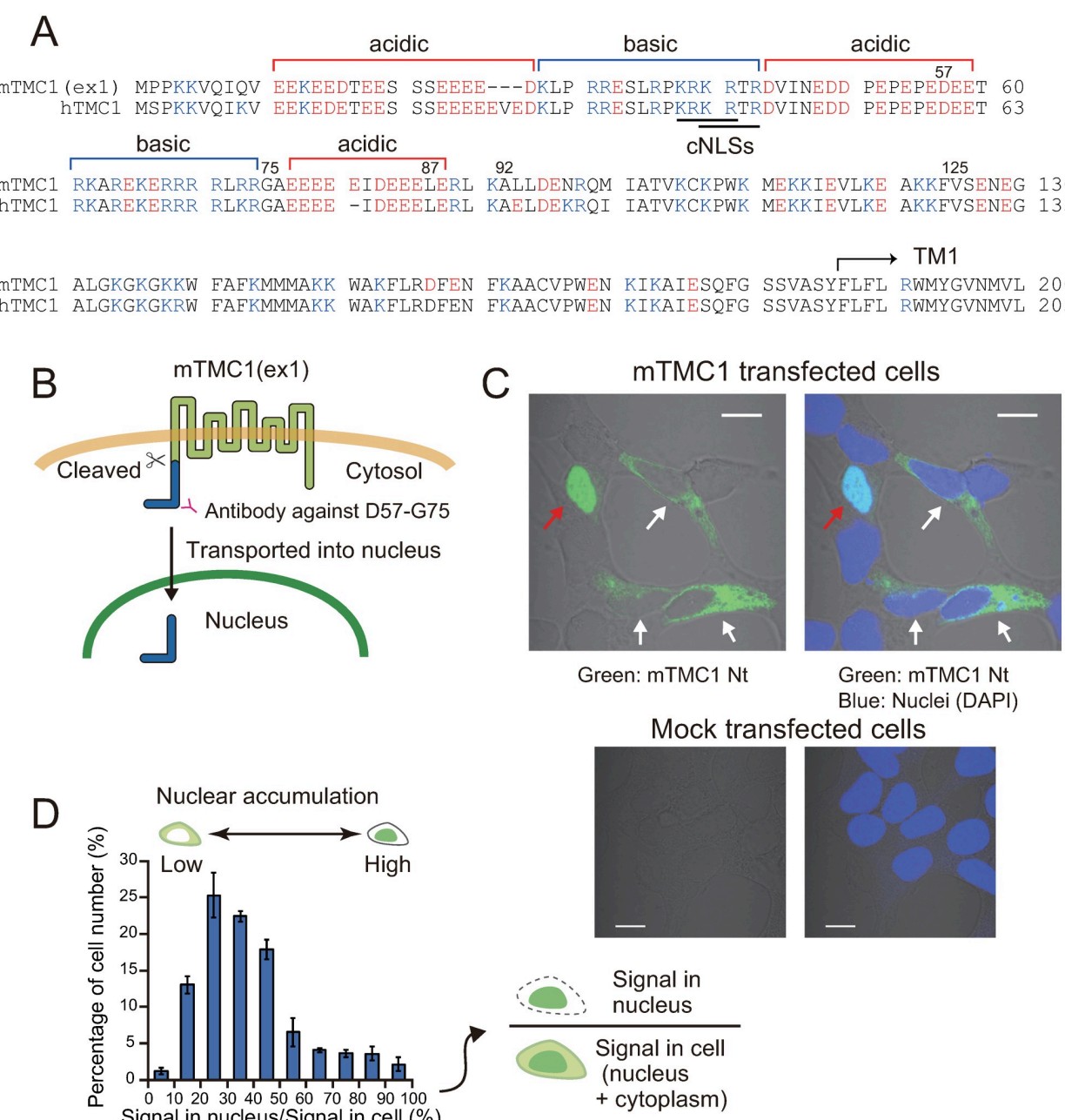

**Fig 1. The N-terminal (Nt) region of heterologously-expressed mTMC1 accumulated in nuclei of a small population of HEK293 cells.** (A) Amino acid sequences of the N-terminal (Nt) regions of mouse TMC1 (mTMC1) and human TMC1 (hTMC1). cNLSs: classical nuclear localization signals. TM1: first transmembrane region. (B) A membrane topology of mTMC1 and the position of the antigen (from 57th aspartate to 75th glycine, D57-G75) of the mTMC1 antibody are shown. Appearance of the Nt region of mTMC1 in a nucleus implies its cleavage and transport into the nucleus. (C) Images of confocal microscopy. Upper two figures are images of mTMC1 transfected cells and lower two figures are images of mock transfected cells. The left figure is a merged image of a green-channel image and a bright-field image, and a blue-channel image was also merged in the right figure. Green signals show the staining of the Nt region of mTMC1, and blue signals show nuclear staining of DAPI. A red arrow indicates a cell where the Nt region of mTMC1 accumulated in the nucleus. White arrows indicate cells where the Nt region of mTMC1 was localized in the cytoplasm (i.e. a normal distribution of heterologously-expressed TMC1). White scale bars indicate 10 μm. (D) A histogram of percentage of cell number, which shows how many cells highly/poorly accumulate the Nt region of mTMC1 in their nuclei. The horizontal axis shows percentage of the signal intensity (green fluorescence of Alexa Fluor 488 i.e. the staining of the Nt region of mTMC1) in the nucleus to that in the whole cell area of each cell. Each class interval of the signal percentage is 10%. Shown are mean ± s.e.m. (n = 4).

When mTMC1 was heterologously-expressed in HEK293 cells, we noticed the Nt region of mTMC1 appeared in the nuclei of some cells. Transmembrane proteins usually cannot enter the nuclei unless their partial fragments are cleaved and released because the entire transmembrane proteins span the cell membrane. Therefore, in this study, we examined whether the Nt region of heterologously-expressed mTMC1 was cleaved. We also identified amino acid residues which were required for the cleavage and evaluated whether cNLSs were involved in the nuclear transportation of the Nt fragment by site-directed mutagenesis. Moreover, we compared and found a similarity between the amino acid sequence of the Nt fragment of mTMC1 and those of basic-region leucine zipper (bZIP) transcription factors. Finally, we analyzed the alteration in transcription by the overexpressed Nt fragment in HEK293 cells by next generation sequencing (NGS) and revealed that the influence on the transcription by the Nt fragment was limited at least in HEK293 cells.

## Materials and methods

### Animal ethics approval

All animal experiments were performed in accordance with guidelines and protocols approved by the Institutional Animal Care and Use Committee, Graduate School of Veterinary Medicine, Hokkaido University (Protocol numbers: 14–0060, 19–0024). Mice were euthanized by $CO_2$ inhalation following the approved procedure.

### Plasmids and cell transfection

We used *mTmc1ex1* cDNA which was cloned from cochleas of male C57BL/6N mice in pcDNA3.1(+) vector (Takara Bio, Otsu, Japan), mammalian expression vectors, as reported previously [23]. Mouse *Tmc2* cDNA was amplified from a cDNA library of dorsal root ganglia of a C57BL/6J male mouse and cloned into pEGFPC1 vector (Takara Bio). The mutants of *mTmc1* were made by site-directed mutagenesis using the PrimeSTAR Mutagenesis Basal kit (Takara Bio). The mutations were verified by sequencing. In some plasmids, 3×Flag tag was added at the N-terminus of mTMC1 using In-Fusion HD cloning kit (Takara Bio). For the fluorescence-activated cell sorting (FACS), the cDNA encoding the Nt partial protein of mTMC1 (M1-E127) was subcloned into pIRES2-EGFP, a bicistronic expression vector which expresses both the product of inserted cDNA and enhanced green fluorescent protein (EGFP) as two separated proteins.

HEK293 cells and HEK293T cells were obtained from the RIKEN BioResource Research Center through the National Bio-Resource Project of the MEXT, Japan. HEK293 and HEK293T cells were cultured in Dulbecco's modified Eagle's medium (Sigma-aldrich, St. Louis, MO, USA) supplemented with 10% fetal bovine serum (Cytiva, Tokyo, Japan) and penicillin/streptomycin (100 U/ml and 100 μg/ml, respectively, FUJIFILM Wako Pure Chemical Corporation, Osaka, Japan) at 37˚C in a 5% $CO_2$ incubator. Cells were transiently transfected with plasmids using TransIT-293 Transfection Reagent (Takara Bio). Thirty to forty-eight hours after the transfection, the cells were used for experiments.

### Immunocytochemistry

The transfected cells on a coverslip were fixed with 4% paraformaldehyde and permeabilized with 0.1% TritonX-100. Non-specific staining was blocked by incubations with image-iT FX signal enhancer (Thermo Fisher Scientific, Waltham, MA, USA) and with Dulbecco's PBS (phosphate buffered saline) containing 2% bovine serum albumin and 5% normal goat serum. The primary antibody was a custom-made anti-mTMC1 Nt region polyclonal rabbit antibody

(antigen: DEETRKAREKERRRRLRRG, D57-G75, Sigma-Aldrich). The secondary antibody was Alexa 488 or 555-conjugated goat anti-rabbit IgG F(ab')2 fragment (Thermo Fisher Scientific). Nuclei were stained with DAPI (4',6-diamidino-2-phenylindole). After washings, the coverslips were mounted with Prolong Diamond (Thermo Fisher Scientific). Fluorescence was observed using a confocal microscope (LSM 700, Carl Zeiss, Jena, Germany).

In order to evaluate the nuclear accumulation of mTMC1 Nt region, the signals of fluorescence of Alexa 488 in a nucleus and in a whole cell area of each cell were measured using Zen3.1 blue edition (Carl Zeiss). The regions of a nucleus and a whole cell area were outlined based on the edges of signals of DAPI and the lines of the cell periphery observed under the brightfield images, respectively. The nuclear accumulation of the signal was evaluated by a proportion of the signal in a nucleus to the signal in a whole cell. About one hundred cells were measured from a coverslip. A histogram of nuclear accumulation (x-axis) versus cell number (y-axis) was made. Measurements were repeated from several different coverslips. Means and standard errors of the data in each class interval were calculated.

## Western blotting

The transfected cells were scraped in an ice-cold lysis buffer (150 mM NaCl, 2 mM EDTA (ethylenediaminetetraacetic acid)·2Na, 10 mM HEPES (2-[4-(2-Hydroxyethyl)-1-piperazinyl] ethanesulfonic acid) (pH = 7.4), 1% Triton X-100, and a protease inhibitor cocktail (1%, P8340, Sigma-Aldrich)) and lysed by sonication. The lysates were centrifuged at 14,000 ×g for 10 min at 4°C. The supernatants were mixed with equal volume of urea sample buffer (8 M urea, 5% SDS, 0.1% bromophenol blue, 0.2 M Tris·Cl (pH = 6.8), and 0.1 M DTT (dithiothreitol) [24]) and heated at 37°C for 10 min. This treatment dissociated mTMC1 protein aggregates and increased the detection of monomeric mTMC1 proteins as described previously [23]. Proteins (20 μg) were separated by SDS-PAGE and transferred to a PVDF membrane. After blocking with skim milk, the membranes were incubated with a commercially available anti-mTMC1 rabbit antibody (ab199949, abcam, Cambridge, UK. Its antigen is somewhere in the first 106 amino acid residues of mTMC1), anti-GFP rabbit polyclonal antibody (E-AB-20050, Elabscience, Houston, TX, USA), anti-HA tag mouse monoclonal antibody (M180-3S, MBL, Tokyo, Japan), or an anti-DYKDDDDK (= anti-Flag) tag mouse monoclonal antibody (FUJIFILM Wako Pure Chemical Corporation). The membrane was incubated with an HRP-conjugated anti-rabbit IgG secondary donkey antibody or HRP-conjugated anti-mouse IgG secondary donkey antibody (Cytiva). β-actin was detected by HRP-conjugated anti-β-actin mouse antibody (Proteintech, Rosemont, IL, USA). At least three samples of mTMC1 were repeatedly detected in each experiment. The chemiluminescent signals were produced using Immobilon Forte Western HRP substrate (Merk, Burlington, MA, USA), detected using a single-lens reflex camera (EOS kiss x7, Canon, Tokyo, Japan), and measured using ImageJ [25].

## Fluorescence-activated cell sorting (FACS)

To sort the HEK293 cells expressing the mTMC1 Nt partial protein, the cells which were transfected with pIRES2-EGFP vector (with or without cDNA of mTMC1 Nt partial protein, M1-E127) were sorted by FACSAria II Cell Sorter (BD Biosciences, San Jose, CA, USA) based on the fluorescence of EGFP. The transfected cells were dissociated using TrypLE Express (Thermo Fisher Scientific) and the cell suspension was passed through a 40-μm cell strainer to obtain single cells. As a negative control, mock-transfected (TE buffer) cells were loaded in order to set the conditions which remove debris, doublets, and EGFP negative cells. EGFP-positive (+) cells were sorted from the cells which were transfected with pIRES2-EGFP vector (with or without cDNA of mTMC1 Nt partial protein, M1-E127) and used for mRNA extraction.

## RNA-seq (NGS)

The transcriptomes of the sorted EGFP-positive (+) cells (the empty vector-transfected cells and the mTMC1 Nt+ cells) were analyzed using Ion Proton sequencing system (Thermo Fisher Scientific). RNA was extracted from the sorted EGFP+ cells using NucleoSpin RNA (Takara Bio). The integrity of the RNA samples was confirmed by measuring RNA Integrity Number equivalent (RINe) using 2200 tapestation (Agilent Technologies, Santa Clara, CA, USA). The RINes of all samples were more than 9.7. mRNA was purified from the RNA samples using Magnospher UltraPure mRNA Purification Kit (Takara Bio). cDNA libraries were prepared using Ion Total RNA-Seq Kit v2 (Thermo Fisher Scientific). The cDNA libraries were checked and quantified using Agilent High Sensitivity DNA kit on Agilent 2100 Bioanalyzer System (Agilent, Santa Clara, CA, USA). The cDNA libraries were amplified by emulsion PCR using Ion PI Hi-Q OT2 200 kit (Thermo Fisher Scientific) and Ion OneTouch2 Instrument (Thermo Fisher Scientific) and condensed using Ion OneTouch ES (Thermo Fisher Scientific). Sequencing was conducted using Ion PI Hi-Q Sequencing 200 Kit, Ion PI Chip v3 kit, and Ion Proton sequencer (Thermo Fisher Scientific). Four barcoded samples were sequenced per one chip. Read numbers were between 17 million to 28 million per sample.

After trimming, the obtained reads were mapped to the reference human genome GRCh38, which was obtained from Ensemble [26], using CLC Genomics Workbench 9 (QIAGEN, Venlo, Netherlands). The annotation data of gene and cDNA used were GRCh38.97. Expression levels of transcripts were estimated as total counts, which were used for statistical analyses, and RPKM (Reads per kilo base per million mapped reads). A statistical significance test between the two groups (the empty vector-transfected cells and the mTMC1 Nt+ cells) was conducted using Empirical Analysis of DEGs tool. FDR (False Discovery Rate) corrected $p$ values were calculated and a $p$ value less than 0.05 was considered significant. From among the genes with more than 1.0 RPKM, we collected genes (total 16 genes) the expressions of which were significantly different by more than two-fold between the mock-transfected cells (Vector) and mTMC1 Nt+ cells (mTMC1 Nt).

## Real-time RT-PCR

Among the sixteen genes, the expressions of *AURKC*, *NPC1L1*, and *TAGLN* in the sorted EGFP+ cells were re-evaluated by real-time PCR using SYBR Green method (PowerUP SYBR Green Master Mix and QuantStudio 12K Flex (Thermo Fisher Scientific)). The three genes were chosen due to their relatively high expressions and large differences between the mock-transfected cells and mTMC1 Nt+ cells. cDNA was synthesized using TAKARA PrimeScript RT master Mix (Perfect Real Time) (Takara Bio). *ACTB* was used as a reference gene. The primers used are listed in Table 1. Specific amplifications were confirmed by each single peak in melting curve analyses. Temperature and timing profiles were: 50˚C/2 min; 95˚C/2 min; and 44 cycles of 95˚C/1 sec, 60˚C/30 sec. Data were analyzed based on the ΔΔCt calculation method [27].

**Table 1. Primers used for real-time PCR.**

| Target gene | Forward primer sequence | Reverse primer sequence |
|---|---|---|
| *AURKC* | CAGGGGTGAGGTGAAGATTGC | TCATTTCTGGCGGCAAGTAG |
| *NPC1L1* | CATCTCTATGGGAAGTGCG | AGCAGAGTGATCAGGAGGTTG |
| *TAGLN* | AATGATGGGCACTACCGTGG | GGCCAATGACATGCTTTCCC |
| *ACTNB* | CACAGAGCCTCGCCTTTG | GCGGCGATATCATCATCC |

## Results

### Accumulation of the N-terminal region of heterologously-expressed mTMC1 in nuclei

As we previously reported [23], and others also confirmed using *hTMC1* [28], among the two major splice variants for *mTmc1* mRNA, *mTmc1ex1* is the splice variant which can translate the protein of mTMC1. Therefore, we used *mTmc1ex1* as the cDNA of *mTmc1* in this study.

We produced an antibody against the cytosolic Nt region (antigen was from 57th aspartate to 75th glycine. Fig 1A and 1B) and used it for the detection of the mTMC1 which was heterologously expressed in HEK293 cells. Although this was an unexpected result, the Nt region of mTMC1 was detected mainly in the nuclei in some transfected cells. (As an example, such a cell is indicated by a red arrow in Fig 1C) Mock transfected cells did not show staining (Fig 1C). In order to analyze how many transfected cells accumulated the Nt region of mTMC1 in their nuclei, we analyzed the percentage of signals of fluorescence in the nucleus to the total signals in the cell and counted the cell number in each class interval (10%) to produce a histogram (Fig 1D). About 13% of transfected cells held more than 60% of the signals of the fluorescence in their nuclei. In other words, about 13% of transfected cells accumulated the Nt region of mTMC1 in their nuclei. On the other hand, Nt region of mTMC2 did not show such an accumulation in the nuclei (S1 Fig).

Because the transmembrane proteins span the cell membrane, they usually cannot enter the nuclei unless their partial fragments are cleaved and released. Therefore, we presumed that the Nt region of mTMC1 was cleaved and transported into the nuclei (as illustrated in Fig 1B) and examined whether the cleaved Nt fragment was detected by western blotting.

### At least a fragment of Nt region of mTMC1 was cleaved from mTMC1

Firstly, we sought to detect the cleaved Nt fragment of non-tagged wild-type (WT) mTMC1 by western blotting. Because the custom-made antibody did not work in western blotting, we used a commercially-available anti-mTMC1 Nt region antibody (ab199949, abcam, Cambridge, UK). Its antigen is somewhere in the first 106 amino acid residues of mTMC1. By using this antibody, at least a fragment of the Nt region of mTMC1 was detected in addition to the full-length band (Fig 2A). Other faint bands (e.g. the lower band around 17 kDa) were also observed in addition to the main fragment band. However, we focused on the main fragment in this study as other bands were too faint to conduct further analyses. Additionally, we also confirmed that the Nt region of mTMC1 was detected in the nuclei of some HEK293 cells using the commercially-available anti-mTMC1 Nt region antibody (S2 Fig).

In order to confirm that there are no more shorter fragments which were undetected by the anti-mTMC1 Nt region antibody, N-terminally 3×Flag-tagged mTMC1 was expressed and its Nt region was detected using an anti-Flag tag antibody. The fragment was detected although the molecular weight of the fragment was increased due to the addition of 3×Flag tag, and no shorter band was observed (Fig 2B).

If the Nt fragment was released by a cleavage, its remainder also must be detected. To examine it, C-terminally HA-tagged mTMC1 was expressed and the Ct side of mTMC1 was detected using an anti-HA antibody. As assumed, a shorter band corresponding to the remainder of the Nt fragment was detected in addition to the full-length band (Fig 2C). The molecular weight of the remainder band was smaller than that of the full-length band by the amount of that of the Nt fragment. The existence of the Ct remainder indicates that the Nt fragment was neither produced by an unexpected arrest of transcription of mRNA nor by that of translation of protein. Moreover, the existence of the Ct remainder also indicates that the Nt fragment was

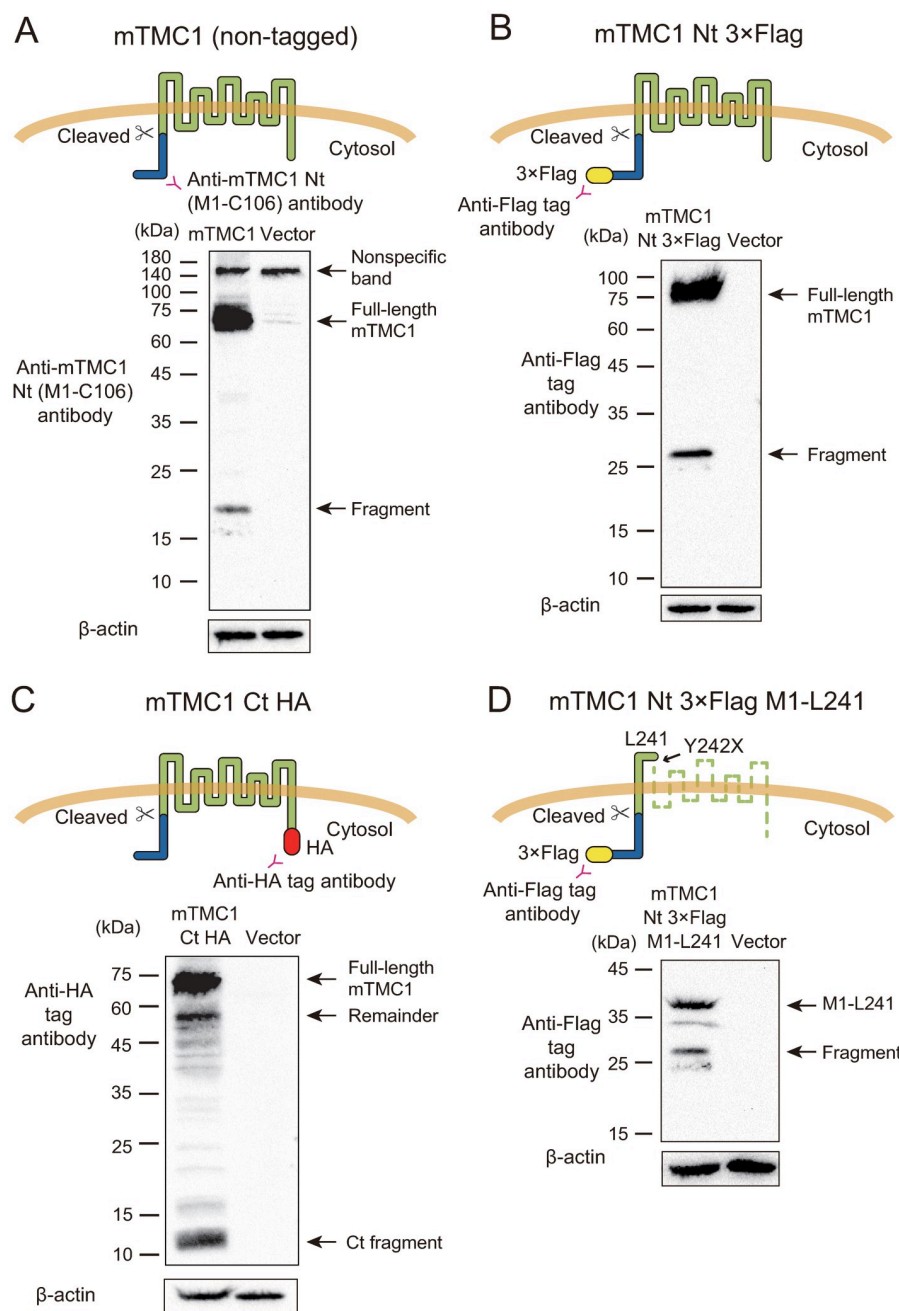

**Fig 2. A cleaved Nt fragment of mTMC1 was detected.** Western blot analyses of (A) non-tagged mTMC1, (B) N-terminally 3×Flag-tagged mTMC1, (C) C-terminally HA-tagged mTMC1, and (D) N-terminally 3×Flag-tagged partial protein of mTMC1 from M1 in the cytosolic region to L241 in the first loop (M1-L241). Upper figures illustrate the constructs which were expressed and the positions of the antigens of the antibodies used. Representative data of western blot are shown below. The lysates of the cells expressing each construct and those of the mock-transfected cells (Vector) were loaded. Thick nonspecific bands between 140 kDa and 180 kDa were observed and labeled. The other nonspecific bands near the full-length mTMC1 were faintly observed in the vector lane in (A) when the anti-mTMC1 Nt region antibody was used as reported previously [23]. The observed Nt fragment was indicated by the allow with a label "Fragment" in (A, B and D). The remainder of mTMC1 was labeled with "Remainder" in (C). As a loading control, β-actin was blotted.

not the remainder by the degradation from the C-terminus by the proteasome [29]. Therefore, this result shows that the Nt fragment was released by a cleavage. Additionally, a short Ct fragment between 10 kDa and 15 kDa was also detected (Fig 2C). When the full-length mTMC1 was separated using a lower percent acrylamide gel, the multiple bands were observed (S3 Fig). These results indicate that Ct region was also released from mTMC1. However, we did not investigate the Ct fragment in this study as our interest was in the Nt region.

In order to examine whether or not the full length of mTMC1 is necessary for the cleavage, a 3×Flag-tagged Nt partial protein of mTMC1 from 1$^{st}$ methionine to 241$^{st}$ leucine (M1-L241), which has a transmembrane region, was expressed. The Nt fragment was also released from the Nt partial protein of mTMC1 in a similar fashion with the full-length mTMC1 (Fig 2D). This result indicates that the cleavage of the Nt region of mTMC1 was independent of both the multiple transmembrane regions and the cytosolic Ct region of mTMC1.

## The Nt region of mTMC1 was less cleaved in HEK293T cells

Intriguingly, the cleavage of the Nt region of mTMC1 was observed less in HEK293T cells, a HEK293 derivative which stably expresses the SV40 large T-antigen [30], than in HEK293 cells (Fig 3). Although the full-length non-tagged mTMC1 was detected more in HEK293T cell samples than HEK293 cell samples, only faintly-visible bands of the Nt fragment were observed from HEK293T cell samples (Fig 3A). Although we have not revealed the mechanism of the difference between HEK293 cells and HEK293T cells, this result suggests that the cleavage was less likely due to an artifact during the preparation of western blotting samples (i.e. an insufficient inhibition of proteases) as the same procedure was conducted using HEK293T cells.

## The cleavage depended on specific amino acid residues

If the Nt region of mTMC1 was cleaved by specific proteases, the cleavages would depend on a specific cleavage site (specific amino acid residues which are recognized and cleaved by the protease) [31]. Therefore, we examined where the cleavage site was in the Nt region of mTMC1 by searching for the specific amino acid residues which were required for the cleavage using site-directed mutagenesis.

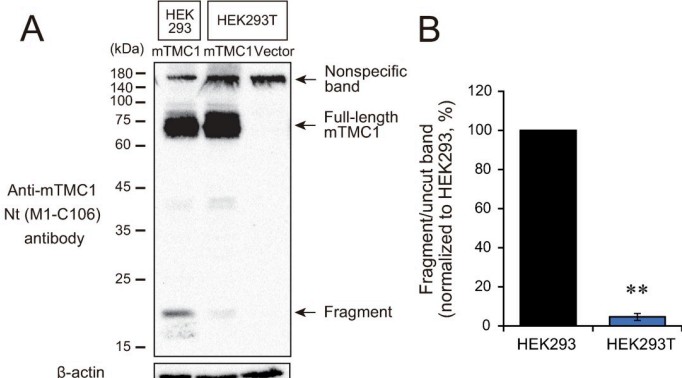

**Fig 3. The Nt region of mTMC1 was hardly cleaved in HEK293T cells.** (A) Non-tagged mTMC1 was expressed in HEK293 cells and HEK293T cells and analyzed by western blotting using the commercially-available anti-mTMC1 Nt antibody. The lysate of mock-transfected HEK293T cells was also loaded (Vector). (B) The ratios of the band intensity of the fragment to that of the uncut band (full-length mTMC1) was quantified and normalized to the value of HEK293 cells (%). The values are mean ± s.e.m., **: $p < 0.01$ with Welch's t-test, n = 5.

From the molecular weight of the fragment, we presumed that the cleavage site would be approximately between 118th leucine (L118) and 131st alanine (A131). We constructed three deletion mutants which lost one-third of the region mentioned above (ΔL118-A121, ΔK122-E127, and ΔN128-A131) from the 3×Flag-tagged Nt partial protein of mTMC1 (M1-L241). From among the three mutations, the deletion of the amino acid residues from 122nd lysine (K122) to 127th glutamate (E127) (ΔK122-E127) eliminated the fragment (Fig 4A). The fragment was also significantly reduced by the deletion of the amino acid residues from L118 to A121 (ΔL118-A121), but the influence was smaller than ΔK122-E127 (Fig 4A). We further investigated which amino acid residues between K122 and E127 were critical for the cleavage producing the fragment by mutating a single amino acid residue to alanine one by one. When either 125th valine (V125) or 126th serine (S126) was mutated to alanine, the fragment was obviously and significantly reduced (Fig 4B). Mutations of surrounding amino acid residues, K123 and E127, also significantly reduced the fragment to a lesser extent (Fig 4B). We also mutated V125 and S126 of full-length mTMC1 to alanine. The mutations (V125A and S126A) also significantly reduced the fragment released from full-length mTMC1 (Fig 4C). These results indicate that V125 and S126 are necessary for the cleavage producing the fragment.

In order to examine whether the amino acid residues (V125 and S126) are near the cleavage site, we compared the size of the fragment released from mTMC1 and an artificially-expressed partial protein which ended at E127 (M1-E127) (Fig 4D). As shown in Fig 4D, the position of the M1-E127 band was slightly lower than but almost identical to that of the released fragment band. Although a few amino acid residues might be different, the fragment was suggested to be produced by the cleavage at the site near E127.

## Nuclear localization signals were necessary for the accumulation of the Nt fragment of mTMC1 in nuclei

We next examined the mechanism underlying the import of the Nt fragment of mTMC1 into the nuclei. In the Nt region of mTMC1, there are two overlapped monopartite classical nuclear localization signals (cNLSs, K-K/R-X-K/R [22]), which are from 38th lysine to 43rd arginine (KRKRTR; KRKR and KRTR) (Figs 1A and 5A). The cNLS is a binding site of importin α, and the complex of importin α and importin β import the cargo proteins which possess the cNLS into the nucleus [22] (Fig 5A). We evaluated the involvement of the cNLSs in the nuclear transport of the Nt fragment by mutating the cNLSs.

When the Nt partial protein (M1-E127) was expressed, almost all of them were accumulated in the nuclei (Fig 5B and 5C). Double-mutations of 39th arginine and 41st arginine to alanine (R39A/R41A) can break both two overlapped cNLSs (Fig 5A). The R39A/R41A mutation of the Nt fragment prevented their accumulation in nuclei and the R39A/R41A mutant was distributed throughout the cell (Fig 5B and 5C). These results indicate that the cNLSs are necessary for the accumulation of the Nt fragment of mTMC1 in nuclei and suggest that importins transport the Nt fragments of mTMC1.

## Comparison between the amino acid sequence of the Nt partial protein of mTMC1 and those of basic-region leucin zipper (bZIP) transcription factors

As the Nt fragment was transported into nuclei, we explored structural similarity between the Nt fragment of mTMC1 and transcription factors. When the amino acid sequence of the Nt partial protein (M1-E127) was analyzed using a secondary structure predicter (Quick2D [32]), the region after 56th glutamate (E56) were predicted to be composed almost entirely of α-

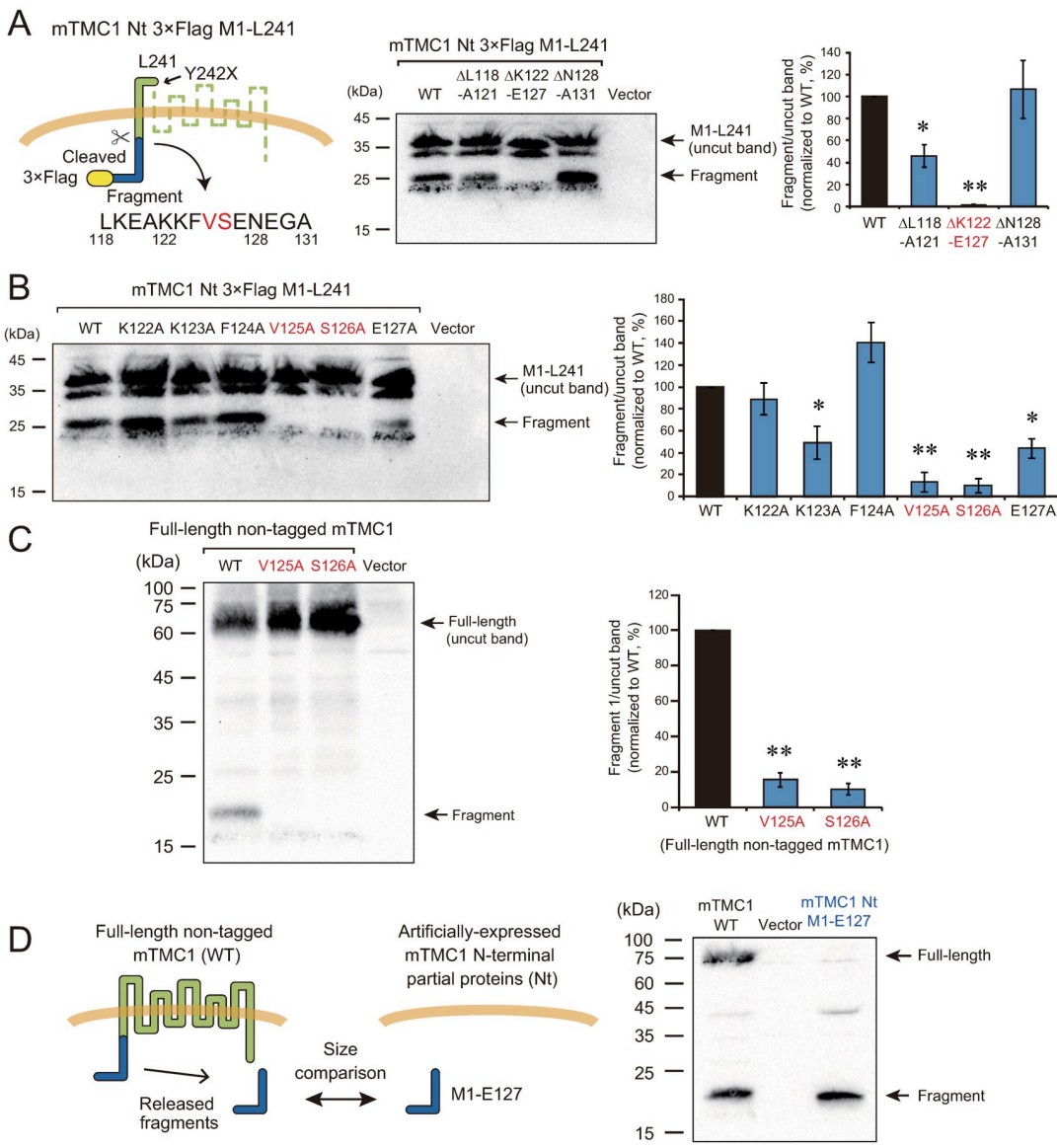

**Fig 4. The 125th valine (V125) and 126th serine (S126) were necessary for the cleavage.** (A) Deletion mutations revealed that amino acid residues from 122nd lysine to 127th glutamate (K122-E127) were necessary for the appearance of the fragment. The positions of mutated amino acid residues are shown on the left. A result of western blot using an anti-Flag antibody is shown in the middle. In the far-left lane, the lysate of the cells expressing Nt 3×Flag-tagged M1-L241 protein was loaded (WT, wild-type). On its right side, the cells were transfected with deletion mutants (ΔL118-A121, ΔK122-E127, and ΔN128-A131) or an empty vector. The arrows indicate the positions of the uncut band of M1-L241 and the band of the fragment. On the right, ratios of the band intensity of the fragment to that of the uncut band were quantified and normalized to the value of WT (%). The values are mean ±s.e.m. *: $p < 0.05$, **: $p < 0.01$ vs. WT with Dunnett's test, n = 4 or 5. (B) Mutations of V125 or S126 to alanine (V125A and S126A) reduced the production of the fragment from M1-L241. Each amino acid residue from 122nd lysine to 127th glutamate was mutated to alanine. A result of western blot using the anti-Flag antibody is shown on the left. The amounts of the fragment were calculated as in (A) and shown on the right. The values are mean ±s.e.m. *: $p < 0.05$, **: $p < 0.01$ vs. WT with Dunnett's test, n = 5. (C) Mutations of V125A or S126A reduced the production of the fragment from the full-length non-tagged mTMC1, also. V125 or S126 was mutated to alanine in the full-length non-tagged mTMC1. A result of western blot using the anti-mTMC1 N-terminal region antibody is shown on the left. The amounts of the fragment were calculated as in (A) and shown on the right. The values are mean ±s.e.m. **: $p < 0.01$ vs. WT with Dunnett's test, n = 6. (D) The design of this experiment is shown on the left. The size of the released fragment was compared with the artificially expressed a mTMC1 Nt partial protein from 1st methionine to 127th glutamate (M1-E127). A result of western blot using anti-mTMC1 Nt region antibody is shown on the right. The band position of the expressed mTMC1 M1-E127 in the far-right lane was almost same with those of the released fragment in the far-left lane.

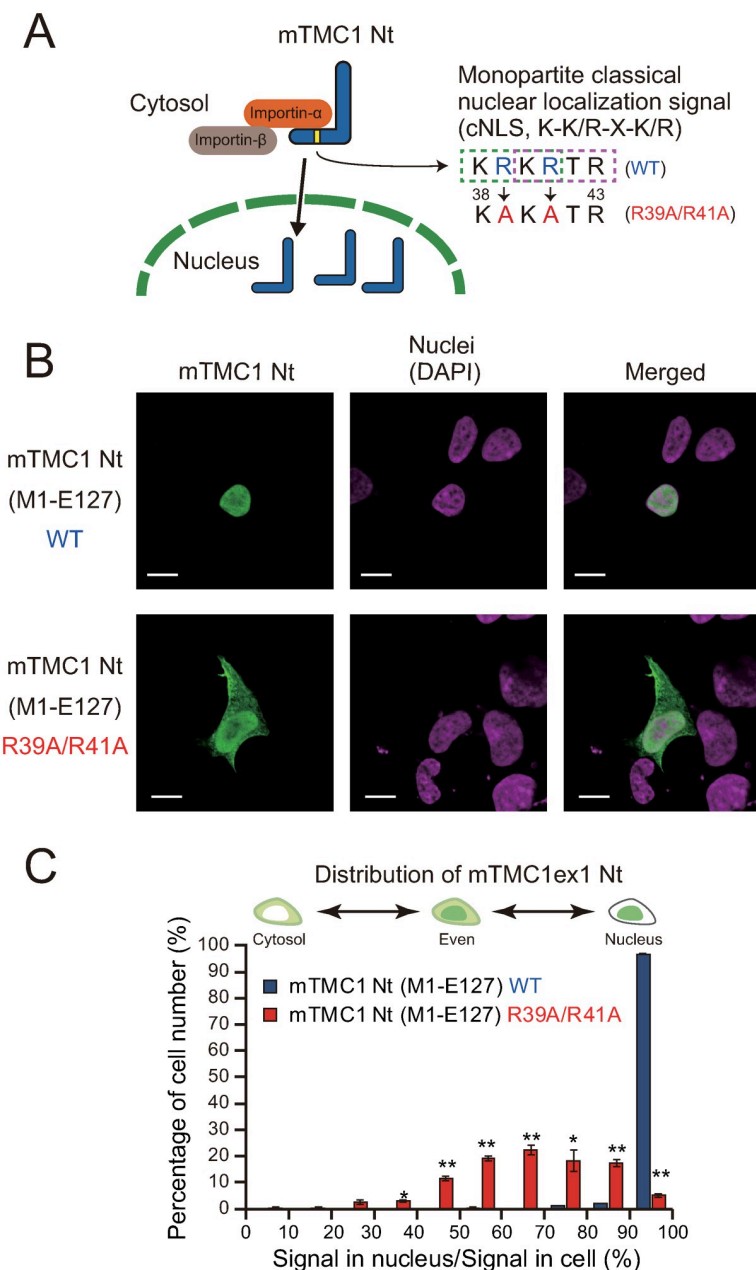

**Fig 5. Classical nuclear localization signals were necessary for the nuclear accumulation of the N-terminal fragment of mTMC1.** (A) The Nt region of mTMC1 contains putative two overlapped monopartite classical nuclear localization signals (cNLSs, K-K/R-X-K/R; K, lysine; R, arginine; X, any residues). cNLS is a binding site of importin α. The double mutations of 39th arginine and 41st arginine to alanine (R39A/R41A) break both the consensus sequences as cNLS. (B) Immunostainings of the mTMC1 N-terminal partial proteins (WT M1-E127 and R39A/R41A mutant). The expressed WT M1-E127 was localized in nuclei, but the R39A/R41A mutant was distributed in the whole intracellular region. Green signals show the staining of the Nt region of mTMC1, and lilac signals show nuclear staining of DAPI. Scale bar = 10 μm. (C) A histogram of percentage of cell number, which shows how many cells highly/poorly accumulate the N-terminal partial protein of mTMC1 (M1-E127) in their nuclei. The horizontal axis shows percentage of the signal intensity in the nucleus to that in the whole area of each cell. Each class interval of the signal percentage is 10%. *: $p < 0.05$, **: $p < 0.01$ vs. WT with Welch's $t$-test, n = 3.

helices. Especially, amino acid sequences from E56 to 105th lysine (K105) and from 77th glutamate (E77) to 115th isoleucine (I115) were predicted to be coiled-coil structures by coiled-coil structure predictors, MARCOIL [33] and PCOILS [34], respectively (Fig 6A). As described in the Introduction section, the Nt region of mTMC1 contains clusters of basic amino acid residues. The first cluster contains cNLSs. The second cluster of basic amino acid residues is within the predicted coiled-coil (Fig 6A). Such a structure reminded us of basic-region leucine zipper (bZIP) transcription factors, which are formed by a dimer of coiled-coils, and the basic region of bZIP works as a DNA binding site [35] (Fig 6B).

The coiled-coils of bZIP are dimerized through repeated hydrophobic residues at "a" and "d" positions of each heptad [35] (Fig 6A and 6B). The "d" positions are typically five consecutive leucines at every seven amino acid residues, for example, as observed in C/EBP, a bZIP transcription factor (Fig 6B and 6C). Interestingly, the Nt partial protein of mTMC1 contains a leucine zipper-like sequence where hydrophobic residues (L: leucine, I; isoleucine, V: valine, P: proline, M: methionine) are repeated at "a" and "d" positions (Fig 6A and 6D). Although there is a polar residue, asparagine (N77), at the "d" position of heptad 2 of mTMC1, there is also an asparagine at the "a" position of heptad 2 of many bZIP transcription factors. The "a" position of heptad 2 of mTMC1 and the "d" position of heptad 2 of all the aligned bZIP transcription factors are leucine. Therefore, the "a" and "d" positions of heptad 2 of mTMC1 are switched in comparison to those of bZIP transcription factors (Fig 6A).

Moreover, many of the "e" and "g" positions of the leucine zipper are occupied by charged amino acid residues (K: lysine, R: arginine, D: aspartate, E: glutamate; Fig 6A–6C), which cover the hydrophobic residues in order to prevent higher order oligomers from forming [35]. Eight amino acid residues out of the ten amino acid residues at the "e" and "g" positions of mTMC1 are charged residues. Therefore, the repeated hydrophobic amino acid residues and the surrounding charged residues of mTMC1 might form a leucin zipper-like structure.

When the leucine zipper-like sequence of mTMC1 was aligned with the leucine zipper sequences of bZIP transcription factors, their basic regions also fit very well ("+" in "Basic region" in Fig 6A). mTMC1 possesses well-conserved, positively-charged amino acid residues (K: lysine, R: arginine) in the basic region although the highly-conserved asparagine (N) is substituted by glutamate (E65) (Fig 6A). Additionally, acidic amino acid residues (D and E) in basic region extension and hinge region are also somewhat conserved although most amino acid residues in the regions are not conserved (Fig 6A).

In this manner, the alignment shows the similarity between the amino acid sequence of the Nt partial protein of mTMC1 and those of bZIP transcription factors. Therefore, we examined whether the Nt fragment of mTMC1 functioned as a transcription factor in HEK293 cells.

## Heterologous expression of the Nt partial protein of mTMC1 scarcely altered gene expression levels

Using NGS, we examined the alteration of gene expression by the overexpression of the Nt partial protein of mTMC1 in HEK293 cells. In order to collect only transfected cells, we used a bicistoronic expression vector (pIRES2-EGFP), which expresses both enhanced green fluorescence protein (EGFP) and the protein encoded by the inserted cDNA. When the vector encoded the Nt partial protein of mTMC1 (M1-E127, corresponding to fragment 1), the transfected cells expressed both EGFP and the Nt partial protein (Fig 7A, Immunocytochemistry). By collecting EGFP positive (+) cells using fluorescence-activated cell sorting (FACS), we can collect only transfected cells and compare the transcriptomes between in the absence and in the presence of the Nt partial protein of mTMC1 (Fig 7A).

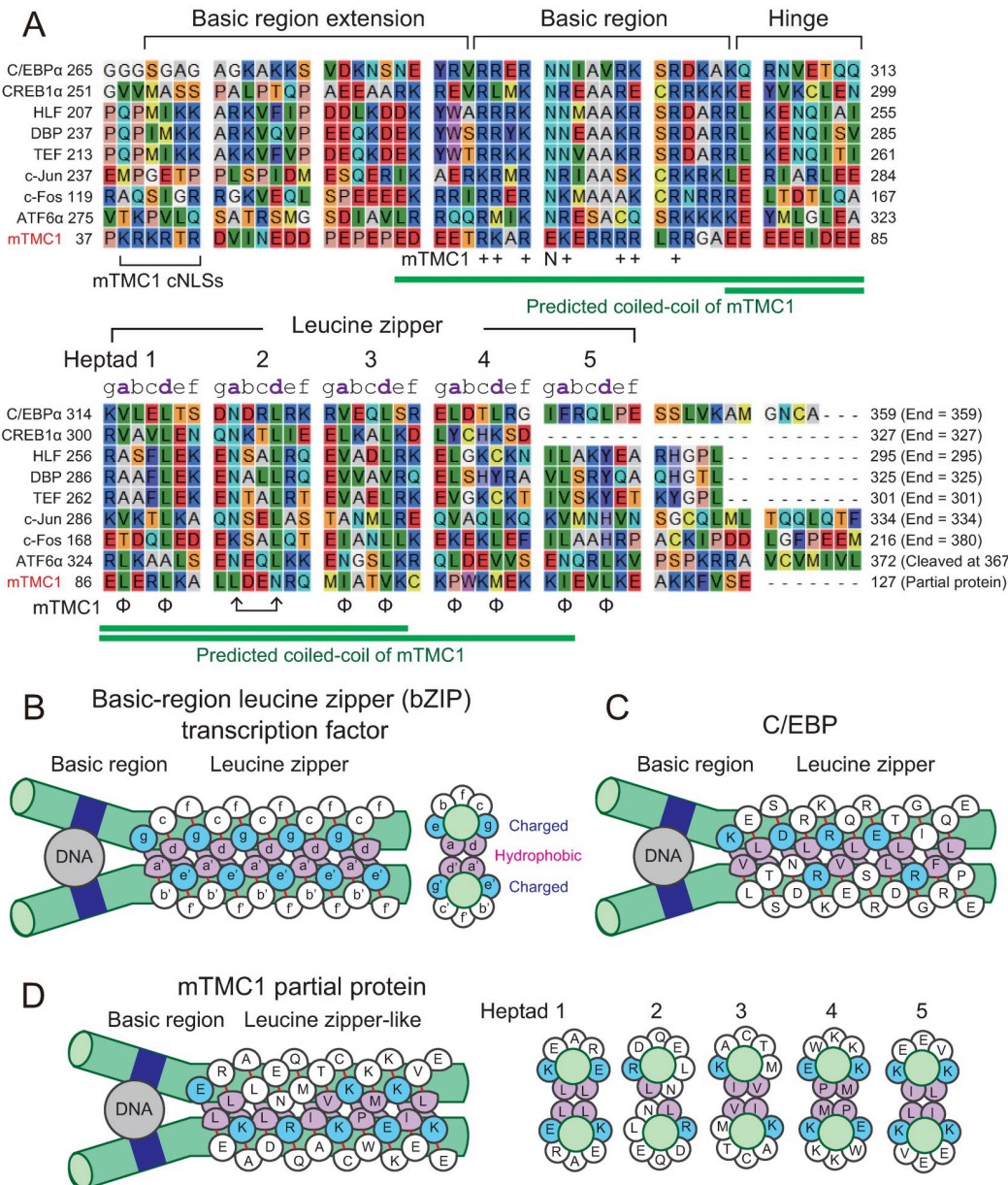

**Fig 6. The amino acid sequence of Nt partial protein of mTMC1 resembles those of basic-region leucine zipper (bZIP) transcription factors.** (A) An alignment of amino acid sequences of bZIP transcription factors of mice (C/EBPα, CREB1α, HLF, DBP, TEF, c-Jun, c-Fos, and ATF6α) and the Nt partial protein (M1-E127) of mTMC1 (amino acid residues from 37th proline to 127th glutamate are shown). Abbreviations are C/EBP, CCAAT/enhancer binding protein; CREB, cyclic AMP responsive element binding protein; HLF, hepatic leukemia factor; DBP, albumin D-box binding protein; TEF, thyrotroph embryonic factor; and ATF, Activating transcription factor. c-Jun and c-Fos are components of the AP-1 site binding protein. The GenBank accession numbers of the amino acid sequences of C/EBPα, CREB1α, HLF, DBP, TEF, c-Jun, c-Fos, and ATF6α are NP_031704.2, NP_598589.2, NP_766151.1, NP_058670.2, NP_059072.1, NP_034721.1, NP_034364.1, and NP_001074773.1, and respectively. The conserved basic amino acid residues which are present in mTMC1 are labeled with plus (+) signs. The hydrophobic residues of mTMC1 at the "a" and "d" positions of the leucine zipper-like sequence are labeled with phi (φ) signs. Green lines indicate the regions of predicted coiled-coils in mTMC1. Each region of bZIP transcription factors was assigned according to a reference [36]. (B) A schematic diagram of a dimer of bZIP transcription factors after [37] (left). The bZIP transcription factors form a dimer (Homodimer or heterodimer). Green cylinders are coiled-coils. The basic region (deep blue) is the basic (positively-charged) residue-rich DNA-binding region. Hydrophobic residues which form the leucine zipper at the "a" and "d" positions are colored purple. Charged residues at the "e" and "g" positions are colored light blue. A cross section of the leucine zipper is shown on the right. (B) A schematic diagram of C/EBP, an example of bZIP transcription factors. (C) An assumed schematic diagram of mTMC1 partial protein (left). Cross sections of five heptads are shown on the right.

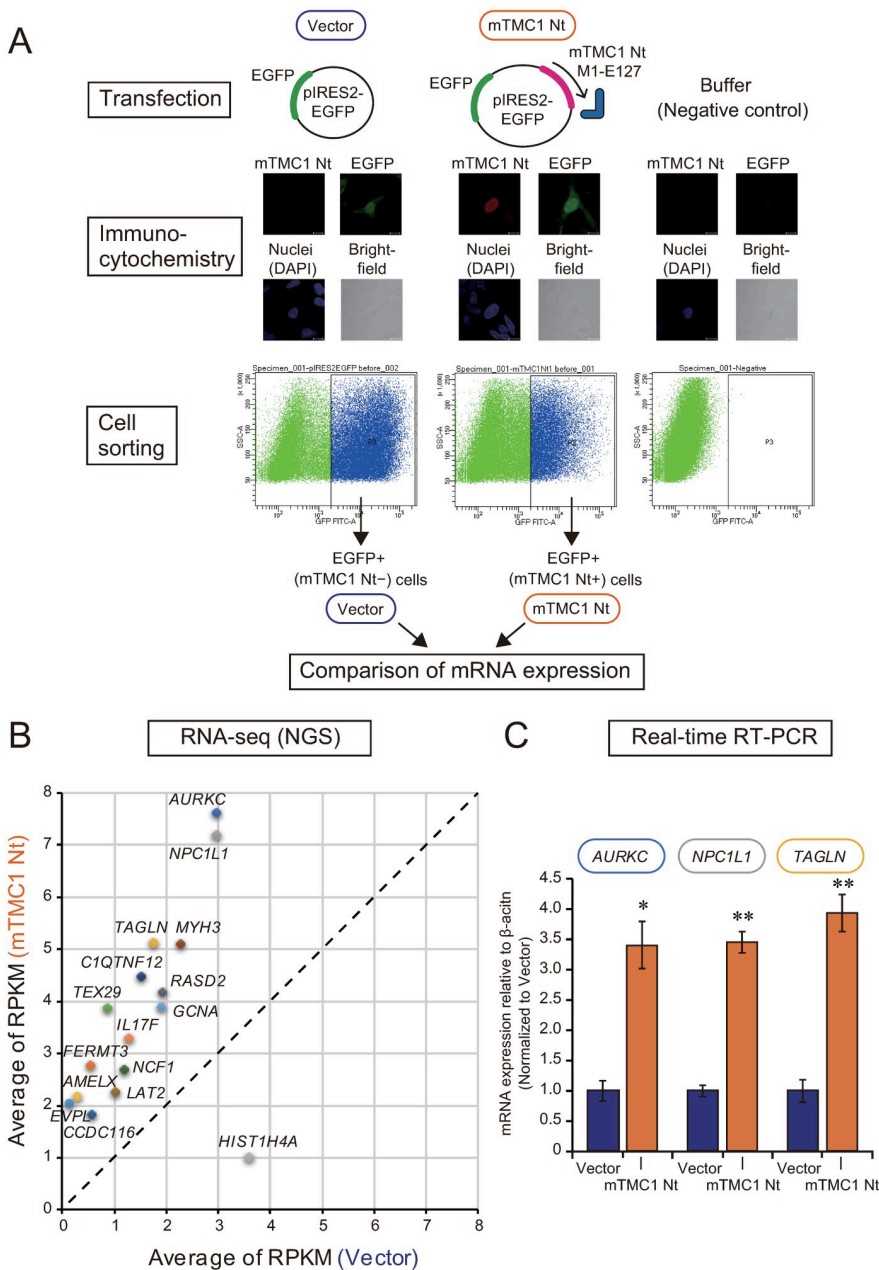

**Fig 7. Heterologous expression of mTMC1 N-terminal partial protein (M1-E127) altered mRNA expression of a few genes in HEK293 cells.** (A) A scheme for analysis of the effect of an expression of the N-terminal partial protein (Nt) of mTMC1 (M1-E127) on transcriptome. The vectors transfected are shown on the top. The results of the immunocytochemistry are shown in the middle. Red signals show the staining of mTMC1 using the anti-mTMC1 antibody (antigen: D57-G75), green signals show the fluorescence of EGFP, and blue signals show nuclear staining of DAPI. Brightfield images are shown on the bottom right. After the cell sorting, which collected the transfected EGFP positive (+) cells (an example of the gate is shown), RNAseq was performed. (B) The RPKM (Reads per kilo base per million mapped reads) of genes the expressions of which were significantly different by more than two-fold between the mock-transfected cells (Vector) and mTMC1 Nt+ cells (mTMC1 Nt) are plotted. The vertical axis and the horizontal axis show average of RPKM of mTMC1Nt expressing cells and mock-transfected cells, respectively (n = 3). (C) Real-time RT-PCR analyses confirmed the increase in the expression of the three genes by the heterologous expression of the mTMC1 N-terminal partial protein (M1-E127). The three genes were selected from among the genes, which were shown to be significantly different by RNA-seq, due to their relatively high RPKM and large difference between two groups. *: $p < 0.05$, **: $p < 0.01$ vs. Vector (n = 3).

As a result of RNA-seq, mRNA expressions of only a few genes (81 genes) among 27,471 humane genes analyzed were shown to be significantly altered by the expression of the Nt partial protein of mTMC1. From among the significantly-changed genes, we collected the genes which showed more than two-fold changes (Fig 7B). The functions, p-values, RPKM, and gene ontologies (GO) of the 16 genes are listed in S1 Table. There are no genes which are obviously related to the function of hair cells in the 16 genes, as far as we know. Three genes (*Aurkc*, *Npc1l1*, and *Tagln*), which showed relatively high RPKM among the 16 genes, were re-evaluated for their expression levels by real-time RT-PCR. The results of the quantitative PCR (qPCR) also showed significant increases in the mRNA expressions of the three genes by the expression of the Nt partial protein of mTMC1 (Fig 7C). However, the increases of the genes' expressions were less than four-fold. These results suggest that the Nt fragment of mTMC1 can modulate the transcription of some genes in HEK293 cells, but the number of affected genes and the extent of enhancement of gene expressions were limited.

## Discussion

This study demonstrated that the Nt region of heterologously-expressed mTMC1 could be cleaved at a specific cleavage site in HEK293 cells and the accumulation of the Nt partial protein in the nucleus was dependent on cNLSs. Additionally, the similarity between the amino acid sequence of the Nt region of mTMC1 and those of bZIP transcription factors was found. However, overexpression of the Nt partial protein of mTMC1 modified the transcription of only a few genes. We discuss the mechanisms and the physiological significance of the observed phenomenon.

One might think that the TMC1 protein was degraded during the sample preparation process such as lysis, but it was unlikely due to the following three reasons. First, the cleavage must have occurred in the cells because the accumulation of the Nt region of mTMC1 in the nuclei was observed in the cells under a confocal microscopy (Fig 1). Second, the cleavage was dependent on the types of cell lines although the sample preparation procedure was the same (Fig 3). Finally, the cleavages required specific amino acid residues (Fig 4), indicating that the cleavage was not mediated by miscellaneous intracellular proteases.

Based on the amino acid residues which were shown to be necessary for the cleavage in this study ($K^{122}KFVSE^{127}$), we searched for the protease which can cleave the sites using *in silico* prediction tools, PROSPER [31] and DeepCleave [38]. However, no cytosolic protease was predicted to cleave these sites. In other words, the site does not contain typical cleavage sites for conventional cytosolic proteases. In order to identify which proteases cleave the site, further experiments such as biochemical and pharmacological analyses are required.

The accumulation of the Nt partial protein into the nucleus was dependent on cNLSs. From this result, it is suggested that importin α binds to the cNLSs and transports the Nt fragment in conjunction with importin β. As importin α and importin β are expressed ubiquitously, if the Nt fragments are released from TMC1 in the plasma membrane in a hair cell, importin α and importin β will probably transport them into the nucleus. There is also a possibility that importin α binds to the cNLSs of the intact (non-cleaved) TMC1 in the hair cells. As TMC1 functions with many interacting proteins [2], importin α might regulate the MET channel function of TMC1. The actual interaction between importin α and TMC1 in hair cells need to be evaluated.

The position of the cleavage which makes the fragment may be ideal to form the leucine zipper-like structure because it ends immediately after the leucine zipper-like structure as many bZIP transcription factors end in similar positions (Fig 6A). Furthermore, bZIP transcription factors containing transmembrane domains such as ATF6, CREB4, OASIS, are known to be

cleaved after the leucine zipper structure by site-2 protease (S2P) [12, 39]. For example, ATF6 is known to be cleaved at the 327[th] cysteine which is about ten amino acid residues after the end of leucine zipper structure [40] (Fig 6A). However, S2P is less likely to be involved in the cleavage of the Nt region of mTMC1 because S2P cleaves proteins within transmembrane regions [41]. Although the protease is probably different, the cleavage of the Nt domain around V125 and S126 may lead to the formation of the bZIP-like structure from mTMC1.

However, the overexpression of Nt partial protein of mTMC1 did not alter expressions of many genes in HEK293 cells. Moreover, the functions of the affected genes are also not specifically related to the hair cell function. These results might be due to the difference of the cell where mTMC1 was expressed. If the Nt fragment of mTMC1 is transported into the nucleus of the hair cells, the Nt fragment might modify expressions of more genes. Additionally, these results might be because the Nt fragment of mTMC1 needs to form a heteromer with other bZIP transcription factors in order to show their physiological function. For example, c-Fos forms a heteromer with c-Jun and functions as a transcription factor, AP-1 [40]. By forming a heteromer with another bZIP transcription factor, the Nt fragment of mTMC1 might function as a transcription factor. Conversely, there is also a possibility that by forming an inactive heteromer with another bZIP transcription factor, the Nt fragment of mTMC1 might indirectly inhibit the function of the bZIP transcription factor. A search for the binding partner of the Nt fragments of mTMC1 might reveal the specific function of the cleaved Nt fragment of mTMC1.

As another possibility of the physiological significance of the cleavage, the cleavage of the Nt region of TMC1 might regulate its channel activity. For example, the remaining channel part of TRPM7 after its cleavage of C-terminal region showed higher ion channel activity than full-length non-cleaved TRPM7 [18]. Furthermore, cleavages of polycystin-1 are required not only for the activation of its signaling function but also for the maturation of polycystin-1 [42]. Similarly, after cleavages of the Nt region of TMC1, the TMC1 function as the MET channel might be affected. Especially, the Nt domain (amino acids 81–130) of mTMC1 was reported to be a binding cite of CIB2 [5, 10] although the other binding cite in the first cytosolic loop of mTMC1 was also reported [11]. CIB2 was shown to be necessary for mTMC1 to function as the MET channel. Therefore, if the Nt region of mTMC1 were to be cleaved to produce the fragment around V125 and S126, the remaining channel part might possess lower MET channel activity through the reduction of the binding to CIB2 in hair cells. Although currently the evaluation of TMC1 in heterologous expression systems are not work, but when it becomes possible, we need to consider the possibility that the cleavage affects the results.

Whether in the case of the transcriptional regulation by the Nt fragment or the regulation of the channel activity by the cleavage, it is quite important to prove that the Nt region of mTMC1 is cleaved in the hair cells *in vivo*. Thus far, although we tried to detect endogenous mTMC1 by western blot using lysates of cochlea and by immunostaining of the organ of Corti, specific signals of mTMC1 were not obtained using the anti-Nt region mTMC1 antibodies. Further research is required to prove that this phenomenon occurs *in vivo*.

## Conclusions

In summary, this study revealed that heterologously-expressed mTMC1 could be cleaved at the Nt region in HEK293 cells. Furthermore, this study not only provided novel information on the Nt region of mTMC1 (the bZip-like structure and cNLSs) but also suggested the cleavage of Nt region might influence the functional analysis of TMC1 by the heterologous-expression system using HEK293 cells. Therefore, this study recommends that HEK293T cells should be used for the functional analysis of TMC1 instead of HEK293 cells in order to avoid

the influence by the cleavage. Further studies will clarify what is the physiological significance of this cleavage and whether it occurs *in vivo*.

## Supporting information

**S1 Fig. mTMC2 did not show the accumulation of Nt region in the nuclei.** (A) Confocal images of N-terminally EGFP-tagged mTMC2 transfected HEK293 cells. The left figure is a merged image of a green-channel image and a bright-field image. A blue-channel image was additionally merged in the right figure. Green signals show the EGFP at the Nt region of mTMC2, and blue signals show the nuclear staining of DAPI. White arrows indicate cells where the Nt region of mTMC2 was localized in the cytoplasm (i.e. a normal distribution of heterologously-expressed mTMC2). White scale bars indicate 10 μm. (B) A histogram of the percentage of cell number, which shows how many cells highly/poorly accumulate the Nt region of mTMC2 in their nuclei. The horizontal axis shows the percentage of the signal intensity (green fluorescence of EGFP at the Nt region of mTMC2) in the nucleus to that in the whole cell area of each cell. Each class interval of the signal percentage is 10%. Shown are mean ± s.e.m. (n = 3).
(PDF)

**S2 Fig. Another antibody against Nt region of mTMC1 detected accumulation of the Nt region in the nuclei.** (A) Confocal images of non-tagged mTMC1 transfected HEK293 cells. The Nt region of mTMC1 was detected by the commercially-available anti-TMC1 Nt (M1-C106) antibody. The left figure is a merged image of a green-channel image and a bright-field image. A blue-channel image was additionally merged in the right figure. Green signals show the staining of the Nt region of mTMC1, and blue signals show nuclear staining of DAPI. Red arrows indicate cells where the Nt region of mTMC1 accumulated in the nuclei. White arrows indicate cells where the Nt region of mTMC1 was localized in the cytoplasm (i.e. a normal distribution of heterologously-expressed mTMC1). White scale bars indicate 10 μm. (B) A histogram of the percentage of cell number, which shows how many cells highly/poorly accumulate the Nt region of mTMC1 in their nuclei. The horizontal axis shows the percentage of the signal intensity (green fluorescence of Alexa Fluor 488, i.e. the staining of the Nt region of mTMC1) in the nucleus to that in the whole cell area of each cell. Each class interval of the signal percentage is 10%. Shown are mean ± s.e.m. (n = 4).
(PDF)

**S3 Fig. The full-length mTMC1 showed multiple bands.** (A) Western blot analyses of non-tagged mTMC1 using an 8% polyacrylamide gel, which was a lower percentage than that used in Fig 2 (12%). The lysates of the cells expressing mTMC1 and those of the mock-transfected cells (Vector) were loaded. mTMC1 was detected by the anti-mTMC1 Nt antibody. The released Nt fragments were out of the gel. As a loading control, β-actin was blotted.
(PDF)

**S4 Fig. Original blots for Fig 2.** Uncropped original blots of western blotting used for Fig 2. Crosses show the lanes which were not used in the Figures. (A) mTMC1 in Fig 2A. (B) β-actin in Fig 2A. (C) β-actin in Fig 2B and 2D. (D) mTMC1 (left) and β-actin (right) in Fig 2C.
(PDF)

**S5 Fig. Original blots for Fig 3.** Uncropped original blots of western blotting used for Fig 3A. Crosses show the lanes which were not used in the Figures. (A) mTMC1 in Fig 3A. (B) β-actin in Fig 3A.
(PDF)

**S6 Fig. Original blots for Fig 4.** Uncropped original blots of western blotting used for Fig 4. Crosses show the lanes which were not used in the Figures. (A) mTMC1 in Fig 4A. (B) mTMC1 in Fig 4B. (C) mTMC1 in Fig 4C. (D) mTMC1 in Fig 4D.
(PDF)

**S7 Fig. Original blots for S3 Fig.** Uncropped original blots of western blotting used for S3 Fig. Crosses show the lanes which were not used in the Figures. (A) mTMC1. (B) β-actin.
(PDF)

**S1 Table. The genes with a significant two-fold change.**
(PDF)

## Author Contributions

**Conceptualization:** Soichiro Yamaguchi, Ken-ichi Otsuguro.

**Data curation:** Soichiro Yamaguchi, Maho Kamino, Maho Hamamura.

**Formal analysis:** Maho Kamino, Maho Hamamura.

**Funding acquisition:** Soichiro Yamaguchi.

**Investigation:** Soichiro Yamaguchi, Maho Kamino, Maho Hamamura.

**Methodology:** Soichiro Yamaguchi.

**Project administration:** Soichiro Yamaguchi.

**Supervision:** Soichiro Yamaguchi, Ken-ichi Otsuguro.

**Visualization:** Soichiro Yamaguchi, Maho Kamino, Maho Hamamura.

**Writing – original draft:** Maho Kamino, Maho Hamamura.

**Writing – review & editing:** Soichiro Yamaguchi, Ken-ichi Otsuguro.

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
