## [Decision Letter · Decision Letter 0]

11 Apr 2023

PONE-D-23-07400The cytosolic N-terminal region of heterologously-expressed transmembrane channel-like protein 1 (TMC1) can be cleaved in HEK293 cellsPLOS ONE

Dear Dr. Yamaguchi,

Thank you for submitting your manuscript to PLOS ONE. After careful consideration, we feel that it has merit but does not fully meet PLOS ONE’s publication criteria as it currently stands. Therefore, we invite you to submit a revised version of the manuscript that addresses the concerns raised by the reviewers. Additional experiments will be needed to convince the reviewers.

We look forward to receiving your revised manuscript.

Kind regards,

Alexander G Obukhov, Ph.D.

Academic Editor

PLOS ONE

Journal Requirements:

   "This research was funded by JSPS KAKENHI Grant Number 16K08066, THE AKIYAMA LIFE SCIENCE FOUNDATION, Takeda Science Foundation, and The Uehara Memorial Foundation to S.Y.This work was the result of using research equipment shared in MEXT Project for promoting public utilization of advanced research infrastructure（Program for supporting introduction of the new sharing system）Grant Number JPMXS0420100619."

  "This research was funded by JSPS KAKENHI Grant Number 16K08066, THE AKIYAMA LIFE SCIENCE FOUNDATION, Takeda Science Foundation, and The Uehara Memorial Foundation to S.Y.

This work was the result of using research equipment shared in MEXT Project for promoting public utilization of advanced research infrastructure（Program for supporting introduction of the new sharing

system）Grant Number JPMXS0420100619. The funders had no role in study design, data collection and analysis, decision to publish, or preparation of the manuscript."

Reviewers' comments:

Reviewer's Responses to Questions

**Comments to the Author**

1. Is the manuscript technically sound, and do the data support the conclusions?

Reviewer #1: Yes

Reviewer #2: Yes

2. Has the statistical analysis been performed appropriately and rigorously? 

Reviewer #1: Yes

Reviewer #2: Yes

3. Have the authors made all data underlying the findings in their manuscript fully available?

Reviewer #1: Yes

Reviewer #2: Yes

4. Is the manuscript presented in an intelligible fashion and written in standard English?

Reviewer #1: Yes

Reviewer #2: Yes

5. Review Comments to the Author

Reviewer #1: The data show that a fragment of Tmc1 that is translocated into the nucleus of HEK293 cells. However, the translated fragment does not interact with other genes.

Overall, the description is well presented but is limited for HEK cells that is in 13% of cells. Moreover, the 16 genes that show the minor effect in HEK cells, are not among the hair cell genes. How is the unusual fragment of Tmc1 play a role in hair cells that is unclear as the data is limited to HEK293 cells.

I strongly suggest submitting a revised version that details the expression in hearing cells that is going beyond limited to HEK293 cells.

Line 49 and other animals (suggest citing the work of Erives, Albert, and Bernd Fritzsch. "A screen for gene paralogies delineating evolutionary branching order of early Metazoa." G3: Genes, Genomes, Genetics 10.2 (2020): 811-826.

Line 67 suggest citing Tmc2-8 that was variably expressed (Erives and Fritzsch, 2020)

Line 466 a citation is needed.

Fig. 1 shows very nicely the nuclear accumulation that is incompletely transfected in the nuclei. Why is transfected in the nuclei??

Fig.3 shows a very limited of HEK293T while 100% are positive for HEK293. The finding is clear the meaning of the differences is unclear.

Fig 5 Eliminate the brightfield and make larger images for the remaining 6 images. Suggest DAPI should be in lilac, not blue.

Reviewer #2: In this study, Yamaguchi and colleagues reported an unrecognized cleavage of N-terminal (Nt) region of TMC1 when heterologously-expressed in HEK293 cells. The authors first observed unexpectedly a nuclear localization of Nt region of overexpressed TMC1 in some population of HEK293 cells. To confirm this might be due to a cleavage in the TMC1 Nt, they carefully conducted extensive western blot assay in combination with mutagenesis and showed that TMC1 is cleaved in Nt around E127 position with V125 and S126 being required for the cleavage. The authors further identified two nuclear localization motifs in the TMC1 Nt region, which are sufficient for translocation of cleaved Nt into the nucleus. The TMC1 Nt share some structural similarity to the bZIP transcription factors. The authors then examined whether the cleaved Nt of TMC1 could affect gene expression profile in HEK293 cells by RNAseq and they found neglectable influence, especially for those genes involved in hearing.

With carefully designed experiments and good controls, this study presented clear evidence to support the authors’ conclusion that TMC1 Nt region is cleaved in heterologous HEK293 cells. This has not been recognized and reported before, so it represents some novelty. My major concern is whether this cleavage also occurs in native hair cells. Since the TMC1 Nt is important for CIB2 binding, it can be speculated that the Nt cleavage would disrupt the interaction between TMC1 and CIB2, and thus functionally affect TMC1 mechanotransduction. I recommend the authors try with immunofluorescence or western blot using hair cells of WT mice or genetic knock-in mice. Some other minor questions are listed below.

1. The authors concluded from Figure 2C that a Ct region of TMC1 also undergoes cleavage, but in the Figure 2A and B no such TMC1 remains after Ct cleavage was detected. Any explanation?

2. Line 315: the 'Fig 6A' and Fig 6B' should be Fig 4D.

3. Does TMC2 also have such cleavage in HEK293 cells?

4. Does expression of M1-L241 fragment of TMC1 prevent cleavage of full-length TMC1? If the cleavage is mediated by some unknown specific protease, the overexpressed M1-L241 fragment might saturate the protease, so the full-length TMC1 could be protected.

5. it is not clear why the antibody against TMC1 D57-G75 generated by the authors does not work in the western blot. Since a commercial TMC1 antibody (from Abcam) was used in the western blot instead, it might be necessary to use the same antibody to assess the localization of Nt in the immunofluorescence experiment.

6. PLOS authors have the option to publish the peer review history of their article (what does this mean?). If published, this will include your full peer review and any attached files.

Reviewer #1: No

Reviewer #2: No

---

## [Author Response · Author response to Decision Letter 0]

23 May 2023

Reviewer #1: The data show that a fragment of Tmc1 that is translocated into the nucleus of HEK293 cells. However, the translated fragment does not interact with other genes.

Overall, the description is well presented but is limited for HEK cells that is in 13% of cells. Moreover, the 16 genes that show the minor effect in HEK cells, are not among the hair cell genes. How is the unusual fragment of Tmc1 play a role in hair cells that is unclear as the data is limited to HEK293 cells.

I strongly suggest submitting a revised version that details the expression in hearing cells that is going beyond limited to HEK293 cells.

Answer 1: Thank you so much for your suggestion. We completely agree with you. As written in lines 470-475 of the original manuscript, we also believed your suggested experiments were quite important and we have already tried them. However, as written in the manuscript, “although we tried to detect endogenous mTMC1 by western blot using lysates of cochlea and by immunostaining of the organ of Corti, specific signals of mTMC1 were not obtained using the anti-Nt region mTMC1 antibodies.” 

 Therefore, we carefully considered what we concluded from our data, and we decided to state that our conclusion of this study is “heterologously-expressed mTMC1 could be cleaved at the Nt region in HEK293 cells.” The conclusion does not go beyond HEK293 cells. Of course, if we can prove that this phenomenon happens in hair cells, the impact of this manuscript will increase significantly. However, even if we cannot prove it, we believe it will not negatively afffect our conclusion in this study. If the conclusion is supported by the data provided, we believe that fulfills the criteria for publication in PLOS ONE. After publication, it is also our hope that not only we but also other researchers who read this paper will examine the translocation of Nt region of TMC1 in hair cells under several pathological conditions.

Comment 1: Line 49 and other animals (suggest citing the work of Erives, Albert, and Bernd Fritzsch. "A screen for gene paralogies delineating evolutionary branching order of early Metazoa." G3: Genes, Genomes, Genetics 10.2 (2020): 811-826.

Answer 2: Thank you for introducing such an intriguing paper to us. We added a sentence, “Not only vertebrates but also metazoa possess TMC-like genes”, in the same paragraph and cited the reference you suggested on line 53 of the revised manuscript.

Comment 2: Line 67 suggest citing Tmc2-8 that was variably expressed (Erives and Fritzsch, 2020)

Answer 3: Thank you. We added the citation on line 67 of the revised manuscript.

Comment 3: Line 466 a citation is needed.

Answer 4: Line 466 is just our assumption. Therefore, there is no citation which should be cited there. The references for the binding of CIB2 were cited on line 469 of the revised manuscript.

Comment 4: Fig. 1 shows very nicely the nuclear accumulation that is incompletely transfected in the nuclei. Why is transfected in the nuclei??

Answer 5: That is not due to the transfection in the nuclei. In fact, the transfected plasmids enter the nuclei during cell division, and mRNA is transcribed from the plasmid in the nuclei. mRNA is transported out from the nuclei, and a protein is translated from the mRNA at the ribosomes on ER. Transmembrane proteins are folded and inserted in ER membrane and transported to the plasma membrane. Therefore, the Nt region of TMC1, a transmembrane protein, should not be observed in the nuclei, and this observation (nuclear accumulation) provided us an idea that the Nt region of mTMC1 was cleaved and transported into the nuclei. 

Comment 5: Fig.3 shows a very limited of HEK293T while 100% are positive for HEK293. The finding is clear the meaning of the differences is unclear.

Answer 6: I’m sorry for causing you confusion. The values (fragment/uncut band) were normalized to the value of HEK293 cells. Therefore, the value of HEK293 cells is 100%. In order to compare how many fragments were released between in HEK293 cells and in HEK293T cells, such normalization was conducted.

Comment 6: Fig 5 Eliminate the brightfield and make larger images for the remaining 6 images. Suggest DAPI should be in lilac, not blue.

Answer 7: Thank you for your nice suggestion. We erased bright field images, changed the color of DAPI, enlarged the pictures, and amended Figure legend for Fig 5.

Reviewer #2: In this study, Yamaguchi and colleagues reported an unrecognized cleavage of N-terminal (Nt) region of TMC1 when heterologously-expressed in HEK293 cells. The authors first observed unexpectedly a nuclear localization of Nt region of overexpressed TMC1 in some population of HEK293 cells. To confirm this might be due to a cleavage in the TMC1 Nt, they carefully conducted extensive western blot assay in combination with mutagenesis and showed that TMC1 is cleaved in Nt around E127 position with V125 and S126 being required for the cleavage. The authors further identified two nuclear localization motifs in the TMC1 Nt region, which are sufficient for translocation of cleaved Nt into the nucleus. The TMC1 Nt share some structural similarity to the bZIP transcription factors. The authors then examined whether the cleaved Nt of TMC1 could affect gene expression profile in HEK293 cells by RNAseq and they found neglectable influence, especially for those genes involved in hearing.

With carefully designed experiments and good controls, this study presented clear evidence to support the authors’ conclusion that TMC1 Nt region is cleaved in heterologous HEK293 cells. This has not been recognized and reported before, so it represents some novelty. My major concern is whether this cleavage also occurs in native hair cells. Since the TMC1 Nt is important for CIB2 binding, it can be speculated that the Nt cleavage would disrupt the interaction between TMC1 and CIB2, and thus functionally affect TMC1 mechanotransduction. I recommend the authors try with immunofluorescence or western blot using hair cells of WT mice or genetic knock-in mice. Some other minor questions are listed below.

Answer 8: Thank you so much for your recommendation. As written in lines 470-475 of the original manuscript, we also believed such experiments were quite important, and we have already tried the experiments which you recommended. However, as written in the manuscript, “although we tried to detect endogenous mTMC1 by western blot using lysates of cochlea and by immunostaining of the organ of Corti, specific signals of mTMC1 were not obtained using the anti-Nt region mTMC1 antibodies.” 

 Therefore, we emphasized that it is not certain whether this phenomena happen in native cells in a paragraph in the Discussion, and so we stated that our conclusion of this study is “heterologously-expressed mTMC1 could be cleaved at the Nt region in HEK293 cells.” Of course, if we can prove that this phenomena happen in hair cells, the impact of this manuscript will increase significantly. However, even if we cannot prove it, we believe it will not negatively affect our conclusion in this study. If the conclusion is supported by the data provided, we believe that fulfills the criteria for publication in PLOS ONE. After publication, it is also our hope that not only we but also other researchers who read this paper will examine the translocation of Nt region of TMC1 in hair cells under several pathological conditions.

1. The authors concluded from Figure 2C that a Ct region of TMC1 also undergoes cleavage, but in the Figure 2A and B no such TMC1 remains after Ct cleavage was detected. Any explanation?

Answer 9: Thank you so much for your careful observation and shrewd discussion. As the Ct fragment was short, the remaining part seemed to be merged with the full-length band in Figures 2A and B. Actually there were (at least) double full-length bands when the bands were separated using lower percentage polyacrylamide gels. We added the picture as Supplementary Figure S3 (S3 Fig) and mentioned this result in lines 277 and 278 of the revised manuscript.

2. Line 315: the 'Fig 6A' and Fig 6B' should be Fig 4D.

Answer 10: Thank you so much. We corrected the wrong numbers.

3. Does TMC2 also have such cleavage in HEK293 cells?

Answer 11: Thank you for your question. Initially, we noticed the accumulation of Nt region of TMC1 using N-terminally EGFP-tagged TMC1 (data not shown). On the other hand, as shown in newly-added Supplementary Figure S1 (S1 Fig), N-terminally EGFP-tagged TMC2 did not show the accumulation of Nt region in nuclei. Therefore, we thought the same phenomena (cleavage and transport into nuclei) did not occur in TMC2 and focused on TMC1. We mentioned this result in lines 243 and 244 of the revised manuscript.

4. Does expression of M1-L241 fragment of TMC1 prevent cleavage of full-length TMC1? If the cleavage is mediated by some unknown specific protease, the overexpressed M1-L241 fragment might saturate the protease, so the full-length TMC1 could be protected.

Answer 12: Thank you for giving us your interesting idea. We think it is not certain whether the overexpressed M1-L241 fragment can saturate the protease because it depends on the activity and the amount of the protease. Therefore, even if the cleavage is mediated by specific protease, co-expression of M1-L241 may fail to protect full-length TMC1. However, we also think the overexpressed M1-L241 fragment might saturate the protease and protect full-length TMC1 if the M1-L241 fragment is cleaved quite more preferentially than full-length TMC1. Your suggested experiment will be useful for the evaluation of the substrate preference of the protease in the next study. We appreciate your suggestion.

5. It is not clear why the antibody against TMC1 D57-G75 generated by the authors does not work in the western blot. Since a commercial TMC1 antibody (from Abcam) was used in the western blot instead, it might be necessary to use the same antibody to assess the localization of Nt in the immunofluorescence experiment.

Answer 13: The reason was because there were relatively more nonspecific bands in the western blotting using the antibody against D57-G75 although it detected TMC1. Following your suggestion, we conducted additional immunocytochemistry experiments using the commercial TMC1 antibody from Abcam. The results (Supplementary Figure S2, S2 Fig) showed that the Nt region of TMC1 accumulated in the nuclei of some HEK293 cells even if the commercial TMC1 antibody was used. We mentioned this result in lines 259 and 261 of the revised manuscript.

---

## [Decision Letter · Decision Letter 1]

2 Jun 2023

The cytosolic N-terminal region of heterologously-expressed transmembrane channel-like protein 1 (TMC1) can be cleaved in HEK293 cells

PONE-D-23-07400R1

Dear Dr. Yamaguchi,

We’re pleased to inform you that your manuscript has been judged scientifically suitable for publication and will be formally accepted for publication once it meets all outstanding technical requirements.

Kind regards,

Alexander G Obukhov, Ph.D.

Academic Editor

PLOS ONE

Reviewers' comments:

Reviewer's Responses to Questions

**Comments to the Author**

1. If the authors have adequately addressed your comments raised in a previous round of review and you feel that this manuscript is now acceptable for publication, you may indicate that here to bypass the “Comments to the Author” section, enter your conflict of interest statement in the “Confidential to Editor” section, and submit your "Accept" recommendation.

Reviewer #1: All comments have been addressed

2. Is the manuscript technically sound, and do the data support the conclusions?

Reviewer #1: Yes

3. Has the statistical analysis been performed appropriately and rigorously? 

Reviewer #1: Yes

4. Have the authors made all data underlying the findings in their manuscript fully available?

Reviewer #1: Yes

5. Is the manuscript presented in an intelligible fashion and written in standard English?

Reviewer #1: Yes

6. Review Comments to the Author

Reviewer #1: (No Response)

7. PLOS authors have the option to publish the peer review history of their article (what does this mean?). If published, this will include your full peer review and any attached files.

Reviewer #1: No

---

## [Editor Report · Acceptance letter]

7 Jun 2023

PONE-D-23-07400R1 

The cytosolic N-terminal region of heterologously-expressed transmembrane channel-like protein 1 (TMC1) can be cleaved in HEK293 cells 

Dear Dr. Yamaguchi:

I'm pleased to inform you that your manuscript has been deemed suitable for publication in PLOS ONE. Congratulations! Your manuscript is now with our production department. 

Kind regards, 

on behalf of

Dr. Alexander G Obukhov 

Academic Editor

PLOS ONE